# The climate in Poland (Central Europe) in the first half of the last millennium, revisited

Rajmund Przybylak[1,2]; Piotr Oliński[2,3]; Marcin Koprowski[2,4]; Elżbieta Szychowska-Krąpiec[5]; Marek Krąpiec[6]; Aleksandra Pospieszyńska[1,2]; Radosław Puchałka[2,4]

[1]Department of Meteorology and Climatology, Faculty of Earth Sciences and Spatial Management, Nicolaus Copernicus University, Toruń, Poland
[2]Centre for Climate Change Research, Nicolaus Copernicus University, Toruń, Poland
[3]Department of Medieval History, Institute of History and Archival Sciences, Faculty of History, Nicolaus Copernicus University in Toruń, Poland
[4]Department of Ecology and Biogeography, Faculty of Biology and Environmental Protection, Nicolaus Copernicus University, Toruń, Poland
[5]Department of Environmental Analysis, Geological Mapping and Economic Geology, Faculty of Geology, Geophysics and Environmental Protection, AGH University of Science and Technology, Kraków, Poland
[6]Department of General Geology and Geotourism, Faculty of Geology, Geophysics and Environmental Protection, AGH University of Science and Technology, Kraków, Poland

*Correspondence*: Rajmund Przybylak (rp11@umk.pl)

**Abstract.** The article presents updated knowledge on climate change in Poland (Central Europe) in the first half of the last millennium (1001–1500). This knowledge is required to delimit the existence, duration and scale of the Medieval Warm Period (MWP, also called the Medieval Climate Anomaly [MCA] here). To this end, it employs all available quantitative climate reconstructions created for Poland in the last two decades and four new reconstructions using three dendrochronological series and an extensive database of historical source data on weather conditions. The growth of conifers in lowland and upland Poland depends on the temperature in the cold season, especially in February and March. All available reconstructions based on dendrochronology data represent this time of the year. Summer temperatures were reconstructed using biological proxies and documentary evidence. The latter, however, is limited to the 15th century only. Winter temperature was used as the proxy for annual temperature proxies instead of the more usual use of summer temperature. The MWP occurred in Poland probably from the late 12th century to the first halves of the 14th or 15th centuries. All the analysed quantitative reconstructions suggest that the MWP in Poland was comparable to or warmer than the mean temperature in the period 1951–2000. The coldest conditions in the entire study period were noted in the first half of the 11th century (both winter and summer) and the second half of the 15th century (only winter). The greatest climate continentality occurred in the 15th century. Good agreement was found between the reconstructions of Poland's climate and many reconstructions available for Europe.

Keywords: Climate change; Medieval times; Documentary evidence; Dendrochronological data; Climate reconstructions; Extreme events

## 1. Introduction

Knowledge about weather and climate in the pre-instrumental period, particularly for the last millennium, is fundamental and necessary to identify past and potential future drivers of climate change (Brönnimann et al., 2019; Brönnimann, 2022; Sieber et al., 2022). The traditional scheme of climate changes in this time that was proposed by Lamb (1965, 1977, 1984), in particular the occurrence of the Medieval Warm Period (MWP, recently called also the Medieval Climate Anomaly, MCA), was recently questioned on a global scale (e.g., Hughes and Diaz, 1994; Brázdil et al., 2005; Diaz et al., 2011); nonetheless, it is still valid for central and western Europe and some other areas mainly in the Northern Hemisphere (Mann et al. 2009). More reliable climate reconstructions for the MCA and the Little Ice Age (LIA) are needed for many areas to obtain better knowledge about spatial temperature patterns. According to Mann et al. (2009), they are still poorly defined.

In the present paper, we focus on climate reconstruction for the area of modern-day Poland in the period 1001–1500, when the anthropogenic fingerprints were small (Hernández-Almeida et al., 2017). According to Niedźwiedź et al. (2015), this period encompasses the MCA (1001–1350) and the Transitional Period (TP, 1351–1500) in the Baltic Sea Basin, which includes Poland. Przybylak (2016), summarising climate knowledge for that time, found that the MCA in Poland began in the 11th century and probably ended in the 14th or early 15th century, and that the LIA started in the mid-16th century.

An increase in knowledge of the Polish climate from the so-called pre-instrumental period is observable in the last several decades. However, the majority of available works cover the entire period since 1500, but more often only short sub-periods thereof (e.g., Przybylak et al., 2005; Przybylak and Pospieszyńska, 2010 and references therein; Opała-Owczarek et al., 2021 and references therein). There are significantly fewer studies dealing with the pre-1500 period compared to the post-1500 period (e.g., Maruszczak, 1988; Sadowski, 1991; Wójcik et al., 2000; Zielski et al., 2010; Przybylak, 2011, 2016; Koprowski et al., 2012; Hernández-Almeida et al., 2015, 2017; Balanzategui et al., 2017; Przybylak et al., 2020; Oliński, 2022).

Przybylak (2016) concluded that "the state of knowledge concerning changes in air temperature in Poland in the period 1001–1500 is very limited […] existing reconstructions being very uncertain. In some periods there are even cases in which opposing trends for the course of air temperature are presented." Significantly worse knowledge is available for other meteorological variables and related phenomena, like droughts (Przybylak et al. 2020) and floods (Ghazi et al., 2023). Such information about the climate of this period is also noted in many different areas of the world. Therefore in recent years, increasing attention has been directed to this period (Vogt et al., 2011; Goosse et al., 2012; Moreno et al., 2012; Pribyl, 2014; Camenisch, 2015; Camenisch et al., 2016).

In this paper we presented an updated knowledge on the Poland's climate in the period 1001–1500 using multiproxy data (mainly documentary evidence and dendrochronological data). For this purpose, a few new quantitative climate reconstructions have been used. Based on documentary evidence, mean 10-year winter (DJF) and summer (JJA) air temperatures for the 15th century were estimated. On the other hand, the dendrochronological data allowed the mean late

winter–early spring temperature to be reconstructed encompassing the period since the 12$^{th}$ century. The newly obtained knowledge allowed us to more precisely described the climate in this period, including the occurrence and scale of warming during the MCA.

## 2. Area, data and methods

### 2.1. Area

The analysis is presented for Poland in its contemporary boundaries (Fig. 1). Eight Natural-Forest Provinces (Zielony and Kliczkowska, 2012) with the location of dendrochronological sites, as well as all historical regions for which documentary evidence are available, are also shown on the map. In the case of historical regions whose boundaries were changeable in the past, some sources were taken into consideration that were written outside of the contemporary Polish territory.

### 2.2. Documentary evidence

The primary type of much of the valuable information about the climate of medieval Poland is narrative sources, e.g., chronicles, annals, memoires and private correspondence. These sources usually describe weather conditions throughout the seasons, mainly in winter and summer. There is also a strong focus on severe droughts, floods and other weather, climate and water extremes (WCWs) like hails, thunderstorms, snowstorms, or sea, lake and river ice. In many cases, they describe the consequences for the economy and society and activities to mitigate their effects.

In our research, we performed a critical analysis of historical sources reporting on climatic phenomena, which allowed us to choose the most reliable "first-hand" sources for the present work.

Another type of source includes administrative and economic sources, i.e., official correspondence or business records containing direct references to WCWs. These also include requests for tax exemption or even financial support related to, for example, flood damage, such as the costs of repairing bridges, flood embankments. In addition to the descriptions of various events, numerical data on sizes or losses of harvests from some of these sources can be processed, and these can then be linked to the economic consequences of the WCWs expressed in quantitative data.

The search for historical sources from this period undertaken by us for the purpose of this work provided significantly more new information than Maruszczak (1991) and Sadowski (1991) used in their reconstructions. We obtained, among other things, many mentions referring to weather from Teutonic correspondence contained in the *Ordens-Brief Archiv* and were able to correct some information from medieval chronicles hitherto either incorrectly dated or uncritically treated as fully credible. It should be emphasised that the first of the aforementioned temperature reconstructions for Poland (Maruszczak, 1991) did not use any historical sources from Poland.

We analysed a total of 813 weather records, of which 145 from the period 1001–1360 (Table S1, Fig. 2) and 668 from the period 1361–1500 (Table S1, Fig. 2). It should be emphasized that there was a significant increase in the number of

available historical written references from the 1360s onwards, and therefore 1360 was chosen as the threshold year for
delimiting the two subperiods. The sources used are listed in Table S2.

We considered it important to list the years from the oldest sub-period analysed in this work (AD 1001–1360) for which reliable historical sources describing weather conditions are available (including those documenting the occurrence of severe winters and hot summers or other seasons) and to make them available in Table S3. Based on these sources, the thermal and pluvial conditions for four seasons were classified jointly using the seven-degree index scale (Pfister et al.,
1994). Examples of typical descriptions of weather in winter and summer based on which we classified thermal and precipitation conditions into seven categories (-3, …, +3) are presented in Table S4. Of those first 360 years, the amount of weather information collected was, as is to be expected, greatest for the 14[th] century (75 items) and least for the 11[th] century (only 13 items) (Fig. 2). We collected the most weather notes for winter (39.3%), undetermined season (24.8%) and autumn (14.5%), and the fewest for spring (9.0%).

For the period 1361–1500, significantly more weather records are available, especially for the last 50 years, when the number ranges around 100 per decade (excluding the decade 1481–90) (Table S1, Fig. 2). As in the previous period, the number of notes was greatest for winter (33.5%). The weather in summer was described with a slightly lower frequency (27.7%).

An indexation of thermal and precipitation conditions was made for the period 1361–1500, similarly as for the
115 previous period. It was based on the information collected from the types of historical sources presented above, was supported by previous catalogues (Walawender, 1932; Girguś and Strupczewski, 1965 database of natural disasters: http://pth.net.pl/projekty/bazy-danych/kleski-elementarne/do -1795) and took into account earlier climate reconstructions for the 15[th] century (Polaczkówna, 1925; Sadowski, 1991; Przybylak, 2011). For both indexation periods, this is the first time for Poland that pluvial conditions have been indexed and that air temperature has been indexed for the transitional seasons
(e.g., spring and autumn).

The new indexation was used to update and significantly supplement the reconstruction of 10-year average winter and summer temperatures for the 15[th] century that were previously presented by Przybylak (2011). Standard methods used commonly in the historical climatology were utilised for this purpose. Pfister et al. (1994) proposed that, for the following index values: i) +3 and -3 should represent anomalies exceeding 2.0 standard deviations (SD) from the mean of the long-
125 term period, ii) +2/-2 and +1/-1 should represent less extreme conditions, i.e. 1.41–2.00SD and 0.7–1.4SD, respectively, iii) 0 (>-0.7SD – <0.7SD) should represent the average climate of the long-term period or missing data. The above-mentioned criteria were slightly modified by us using results of calibration of documentary evidence with air temperature data from Warsaw for the period 1789–1850 done by Sadowski (1991). As a result, the seasons described in this paper by indices from +3 to -3 fulfil the following criteria:

$$+3 \geq m + 1.5SD$$

$$m + 1.0\text{SD} \leq +2 < m + 1.5\text{SD}$$
$$m + 0.5\text{SD} \leq +1 < m + 1.0\text{SD}$$
$$m - 0.5\text{SD} < \quad 0 < m + 0.5\text{SD}$$
$$m - 1.0\text{SD} < -1 \leq m - 0.5\text{SD}$$
$$m - 1.5\text{SD} < -2 \leq m - 1.0\text{SD}$$
$$-3 \leq m - 1.5\text{SD}$$

where $m$ is the long-term mean (for calibration period 1789–1850) air temperature from the Warsaw series and SD is the standard deviation of that series. For more details, see Przybylak et al. (2005). The reader should be reminded of the possible biases related to climate reconstruction based on the documentary evidence such as, for example: i) the number of available sources, their quality and discontinuous structure, ii) the subjectivity of the indexation, iii) weaknesses of the reconstruction method used. A detailed overview of the strong points and the drawbacks of documentary data is presented by Brázdil et al. (2005).

The amount of information available for the period 1361–1400 (Table S1, Fig. 2) does not currently permit a quantitative reconstruction. For summers, too, for the same reason, it was not possible to reconstruct the first and third decades of the 15th century.

A comparative indexation based on historical sources and dendrochronological data was also undertaken that indicated consistencies and disparities in the assessment of individual extreme years.

## 2.3. Dendrochronological data

### 2.3.1. Material and removal of age-related development trends

The regional tree-ring chronologies used in the study were developed by Zielski (1997), Krąpiec (1998), Zielski and Krąpiec (2004) and Szychowska-Krąpiec (2010). The wood samples were taken from forest stands, historic architectural objects and archaeological excavations (Fig. 1). The cores from living trees were taken using incremental borers of 5 mm diameter at a height of ~1.5 m, dried and mounted on wooden holders (Cook and Kairiukstis, 1990). Samples were taken from historic buildings, in the form of 15-mm cores or discs. The measuring paths were cut along 2 or 3 core radii using a dissecting knife. The width of annual rings was measured under microscope with an accuracy of 0.01 mm. Each dated series was checked for measurement errors and missing rings using the COFECHA software (Holmes, 1983). The averaged measurement series of growing trees and historical wood were combined into one regional chronology.

To remove age-related trends from raw grain width measurements, the procedure described in Melvin and Briffa (2008) was used, with processing in RCSigFree 45_v2b software (Cook et al., 2014). Variance was stabilised using an Rbar-weighted method to adjust for changes in sample size (Osborn et al., 1997; Frank et al., 2007) followed by a robust bi-weight mean applied to develop a final indexed chronology (Cook, 1985; Cook and Kairiukstis, 1990).

The current Rbar and Expressed Population Signal (EPS) metrics were computed in 21-year windows with a 10-year overlap. These metrics were used to assess the strength and stability of the common signal between trees over time, and to determine the length of the final chronology used for the reconstruction. Rbar is a measure of the percentage of shared variance between individual runs, with higher Rbar values indicating a stronger common signal (Briffa, 1995). EPS, by contrast, provides estimates of how well a finite number of samples represent a theoretically infinite population (Wigley et al., 1984). EPS values above 0.85 were taken as indicating a period when the ring-width data showed a strong common signal (Table 1).

The sensitivity of the tree-ring chronology of the examined trees to the climate was tested using long-term series of average monthly air temperature values and monthly precipitation sums from Kraków and Toruń (Fig. 1) using the method proposed by Fritts (see Table 1). We confirmed (Table 1) the results presented earlier by Zielski (1997), who reveals statistically significant correlations between annual tree-ring widths (Kuyavia-Pomerania dendrochronology, 1891–1991) and the monthly mean air temperatures from the region, particularly from February and March, but also from January and April. Their values were equal to 0.47, 0.55, 0.26 and 0.18, respectively. This means that, in Poland, the low temperature occurring at the end of winter and at the beginning of spring has a strong negative influence on the width of tree-rings. On the other hand, precipitation has a weaker influence than temperature, though only in June and July is this statistically significant. More information about climate signal in selected trees in Poland is provide in Table 7.1 in Zielski et al. (2010). After identifying the strongest and most time-stable relationship between ring width and the listed meteorological parameters, a transfer function based on linear regression was built using ring-width chronology as the climate predictor. To assess the stability of the linear model, calibration/verification tests were performed at different times depending on the length of the climatic data and chronology (Table 1). The calculations were made using the R software (RCoreTeam, 2022), the dplR package (Bunn, 2008) and treeclim package (Zang and Biondi, 2015).

### 2.3.2. Moon rings, included sapwood

A good indicator of extreme cold thermal conditions during winter is moon rings (MR), also known as "included sapwood", which are found in the wood of European oaks (*Quercus robur* and *Quercus petraea*). They are so called for their presentation as halos in the cross-section of the dark heartwood. The wood of an MR shows an absence or low number of tyloses in the vessels and a lower content of heartwood substances (Dujesiefken and Bauch, 1987; Dzbeński and Krutul, 1994). MR are caused by disturbances in starch management disrupting the heartwood formation process (Dujesiefken and Bauch, 1987) and is caused particularly by cold winters (Bolychevtsev, 1970; Dujesiefken and Liese, 1986; Krąpiec, 1998).

Samples with MRs were identified from a collection of about 2,500 discs and cores from oaks from Holocene alluvial deposits in southern Poland, archaeological excavations and wood from buildings. MR zones were identified on a prepared cross-sectional area under a binocular magnifier. For each site, a correlation diagram of trunks with increments in the MR zone marked was prepared. Since some of the samples come from the lower trunk, and the area of MR is smaller in

this part (Krąpiec, 1999; Dujesiefken and Liese, 1986), the dating of the last increment from the MR zone may differ by about 2–5 years from the actual date of the moon ring's formation.

### 2.3.3. Climate reconstruction pointer years

For climate reconstruction, we employed two approaches. The first involves the pointer years identified for all chronologies, both oak and coniferous trees (Table S5), and the second involves a regression model used for pine from the Kuyavia-Pomerania region and pine and oak from Lesser Poland.

Pointer years (Huber and Giertz-Siebenlist, 1969) were defined as those in which weather conditions cause the vast majority of trees to develop a narrower or wider ring than in the previous year. They appear as a result of various factors, both short-term (e.g., one-night, late-spring frosts) and long-term (e.g., droughts, severe winters, cold spells) (Schweingruber, 1992). Despite the weather conditions of individual pointer years sometimes being hard to explain, their usefulness in climate change research and dendrochronological dating is unquestionable. In this paper, the criterion for

pointer years was that a 90% match be found for a threshold of ten or more trees.

### 2.3.4. Climate reconstruction regression model

Previous research proves that the correlation between Scots pine tree rings and climate in northern (Zielski, 1997; Zielski et al., 2010; Waszak et al., 2021) and southern (Szychowska-Krąpiec, 2010) Poland is strongly dependent on temperature.

Similar relationships were observed for fir from southern Poland (Szychowska-Krąpiec, 2010). Hence, we assumed that the previous findings justify our temperature reconstruction. After the study of the regression model, the calibration period for three sites was chosen as the earlier one (Table 1). The highest correlation was observed for fir from Lesser Poland, with 0.57 for the calibration period and 0.49 for the whole period. Positive RE and CE values for verification intervals indicated that the chosen linear models have good predictive skill suitable for reconstruction. Furthermore, the RMSE test is very

good, with values below 1.

## 3. Results

### 3.1. Documentary evidence

### 3.1.1. 1001–1360

For this period, there exists only one reconstruction of air temperature based on documentary evidence (Sadowski 1991).

The author calculated the frequencies of severe winters and hot summers since the 13[th] century using weather descriptions available mainly in the chronicle *Annales seu cronici incliti regni Poloniae* of Jan Długosz (1415–80). He found 18 severe

winters and 18 hot summers in this time. An exceptionally high decadal value of hot summers (7) occurred in the 1330s. Only in one decade (the 1280s) did as many as three severe winters occurred.

The number of sources for the period 1001–1360 is small and has remained essentially unchanged for many years. The probability of finding new sources is negligible, as also shown by our query. As a result, the number of mentions of weather conditions we have collected is limited (Table S1, Fig. 2). They cannot be used to create a quantitative reconstruction – nor even a full qualitative reconstruction – of changes in climatic conditions. Meanwhile, the scarcity of the available sources requires that their credibility should, all the more, be thoroughly reassessed. Below, we present only selected examples of extreme hydrometeorological events recorded in the sources. However, in the Table S3 we present all such events. The descriptions are decidedly dominated by information on floods and severe winters. Jan Długosz in his *Annales* was the first to mention the weather in Polish lands when describing the year 988. Then, after numerous floods, there was an exceptionally hot summer.

For severe winters, the descriptions indicated rivers freezing, the Baltic Sea freezing, and the prevailing winter conditions in descriptions of military events. Several Western European sources tell, for example, of a severe winter in Lusatia in 1069. These descriptions were associated with a military campaign being waged by the German emperor in that territory. Such weather conditions may also have prevailed in Poland at that time. Western sources also mention the severe winter of 1076/77, during which the Elbe and Vistula rivers froze over. From sources established in Poland lands, we know of the severe winters of 1110/11, 1124/25, 1204/05, 1234/35, 1252/53, 1257/58, 1279/80, 1285/86, 1305/06 and 1322/23. However, not all of these severe winters are entirely certain. Some doubts as to the sources exist regarding the winters of 1234/35, 1252/53, 1257/58 and 1305/06. Meanwhile, it has been erroneously reported that there was a severe winter in Polish lands in 1224/25.

In the period 1001-1360 we have evidence for occurrence of 22 extremely severe and very severe (indices –3 and –2) winters, and only one extremely warm and very warm (3 and 2) summer.

### 3.1.2. 1361–1500

Of the study period, our supplementary query significantly increased the number of historical sources describing weather conditions for only the second half of the 14th century onwards. Thus, the knowledge about the climate of Poland presented here is new, being the most complete and reliable record to date. Particularly valuable is the quantitative reconstruction of the winter and summer temperatures for the 15th century.

The frequency of winters (DJF) and summers (JJA) in Poland that were either extremely warm and wet or extremely cold and dry in the period from 1361 to 1500 is shown in Table 2. It is in line with expectations that sources reported the occurrence of extremely cold and very cold winters with a greater frequency (41) than extremely warm and very warm winters (10). For summers, on the other hand, extremely warm and very warm seasons were about twice as frequent as cold seasons. Large and very large amounts of precipitation were associated with extremely warm and very warm winters,

and were significantly rarer during cold winters. On the other hand, the thermal character of summer (warm, cold) did not differentiate the occurrence of heavy precipitation (Table 2). Time runs of seasonal frequencies of all categories of thermal and humidity extremes in Poland in 10-year periods are shown in Figs 3 and 4. In the study period, but in particularly in the 15[th] century, extremely cold and very cold winters dominated almost all decades (Fig. 3). Such winters were frequent in the 1430s and 1450s (7 cases each) and the 1490s (5). In the first two decades, springs were also very cold. Close to normal thermal conditions in summer were noted in the first 80 years (1361–1440). Later, a decade-to-decade warming of summers accelerated, to reach a maximum in the 1470s (Fig. 3). In the two warmest summer decades (1461–70 and 1471–80) warming was also noted very often in autumn (Fig. 3).

The documentary evidence gives significantly less information about the wetness of the seasons. In line with expectations for Poland, such information dominates for summer and then for winter, but is only rarely available for the transitional seasons (Fig. 4). Very little information was gathered for the 14[th] century. In the 15[th] century, all categories of wet winters dominated, with a maximum in the 1430s (4). This means that the 1430s in Poland were characterised by very cold and snowy winters. Except for last three decades of the 15[th] century, summers were also dominated by extremely wet and very wet conditions, in particular in the 1460s (5 cases). Radical dryness of the air occurred in the 1470s (Fig. 4).

According to the new reconstruction (Fig. 5), average winter temperatures were lowest in the 1450s. Then, the average winter temperature was -6.7 °C and was as much as 5.3 °C lower than in the period 1951–2000 in Poland (-1.4 °C, Kożuchowski and Żmudzka, 2003) and 6.1 °C lower than in 1991–2020 (-0.6 °C, Tomczyk, 2022). These two present mean temperatures for the area of Poland were calculated based on data taken from 45–50 and 40 weather stations, respectively. The next coldest decades in terms of winter temperatures were the 1430s (-5.5 °C) and the 1490s (-4.8 °C). The warmest winters were in the decade 1481–90 (-3.4 °C). In the contemporary climate of Poland, only the coldest winters (e.g., winter 1995/96 with an average value of -5.1 °C, Tomczyk, 2022) reached values similar to the reconstructed 15th-century winter temperatures.

The decadal average summer temperature in Poland in the 15[th] century ranged from 17.8 °C (1431–40) to 19.2 °C (1471–80), as compared to 17.0 °C for 1951–2000 (Kożuchowski and Żmudzka, 2003) and 17.6 °C in 1991–2020 (Tomczyk, 2022). In all decades of the 15[th] century, the decadal average summer temperature was significantly higher than at present, with positive anomalies (relative to 1951–2000) ranging from 0.8 °C to 2.2 °C (Fig. 5). In recent years, however, the summer temperature in Poland has been comparable to that in the warmest decade of the 15[th] century; for example, in the summer of 2019 it was 19.5 °C (Tomczyk, 2022). The presented results also clearly confirm the correctness of the assessment of the Polish climate as continental in this period, as calculated by Sadowski (1991, see his Fig. 6). He showed that the climate of the 15[th] century was the most continental, far more so than the contemporary climate or that of the 13[th] and 14[th] centuries.

## 3.2. Dendrochronology

### 3.2.1. Moon rings (MR)

During the period 1001–1500, MRs, which represent cold winter conditions, were found for years 1141/42, 1314/15, 1328/29 in Nysa; 1186/87 in Czermno (Lublin Province); 1332/33 in Olsztyn; 1397/98 in Gieczno (Łódź); 1422/23 (24?) in Branice near Krakow; 1414/15(16?), 1426/27, 1440/41(?), 1450/51, 1459/60, 1470/71, 1480/81, 1491/92 in Wrocław; 1370/71, 1443/44, 1458/59 in Kraków; 1480/81, 1490/91(92?), 1499/1500 in Kutno; 1408/9, 1453/54 in Domachowo (Greater Poland) and 1381/82 in Węgrów (Mazovia).

The time distribution exhibits a few MRs coinciding across a larger territory simultaneously. These events are limited to the 15th century, when they were found in southern Poland (Wrocław, Kraków) in the 1440s and 1450s, and in Wrocław and Kutno in the early 1480s and 1490s.

### 3.2.2. Pointer years

The range of oak chronologies for Greater Poland, Lower Silesia and Lesser Poland covered the entire analysed period of AD 1001–1500. Sequences of tree rings in specific regional chronologies were used to determine pointer years for Lesser Poland, Greater Poland and Lower Silesia (Table S5 and Figs 6 and 7). For the study period, only two pointer years were found to be common to all three regions – one negative (1401) and one positive (1186). In total, for the years 1001–1500, 34 pointer years were observed in Lower Silesia, 38 in Lesser Poland and 39 in Greater Poland (Figs 6 and 7).

For the pine chronology from Lesser Poland in the study period (1091–1500), 27 pointer years were found (Table S5, Figs 7 and 8), of which, 14 were negative pointer years.

This was fewer than in Lesser Poland's fir chronology (1109–1500), where 47 pointer years were found (Table S5, Figs 7 and 8). The distribution of pointer years is quite uniform between the centuries, except for the early 12th century and the exceptional 13th century, when as many as 21 pointer years were found.

In the case of pine from Kuyavia-Pomerania for the years 1168–1500, 25 pointer years were identified, of which 21 are negative (Table S5, Figs 7 and 8).

### 3.2.3. Regression model

Temperature reconstructions for the winter months based on the three constructed chronologies are presented in Fig. 9 and in a more generalised form in Fig. 10A–C. For the years AD 1091–1500, the average temperature of February–March (Fig. 9b) was reconstructed using the residual pine chronology, whereas for 1109–1500 the average temperature of December–March was reconstructed (Fig. 9c) using the residual fir chronology as a predictor. In both reconstructions, the temperatures in the study period were lower than those recorded today (1951–2000), whether comparing the study period to the anomaly

reconstruction based on the pine or that based on the fir chronology. However, in the reconstruction of average February–March temperatures using the chronology for 1168–1500 (Fig. 9a), warmer periods were observed in the 13[th] and 14[th] centuries, followed by a slight cooling.

## 4. Summary and discussion

To improve our knowledge concerning climate change in Poland from 1001–1500, we constructed three new late-winter–early-spring temperature reconstructions covering the period since the 12[th] century based on tree-ring widths (Figs 9 and 10A–C). In addition, winter and summer temperatures for the 15[th] century were reconstructed based on documentary evidence (Fig. 5). Unfortunately, the scarcity of historical sources (see Fig. 2) for the area of present-day Poland does not allow such reconstructions for the earlier centuries. For the complete synthesis of Poland's climate in the study period and for comparison purposes, we used all available reconstructions in the literature of cold season temperatures based on tree-rings widths (mainly *P. sylvestris*) covering medieval times: Przybylak et al. (2001), Jan–Apr, 1170–1994, Szychowska-Krąpiec (2010), Dec–Mar 1091–2006 and Dec–Mar 1109–2004 [*A. alba*], Koprowski et l.. (2012), Feb–Mar 1168–2000, and Balanzategui et al. (2017), Nov–Apr 1200–2010). See also the recently published inventory of all available dendrochronologies for Poland (Opała-Owczarek et al., 2021 Table 5.2). Information about climate conditions occurring in Poland in the 11[th] century is available only from analysis of biological proxies taken from laminated (varved) sediments in Lake Żabińskie (NE Poland). Quantitative reconstructions of the cold season (Hernández-Almeida et al., 2015) and August temperatures (Hernández-Almeida et al., 2017) are available for the entire millennium (Fig. 10F,G).

The 11[th] century in Poland was the coldest in the entire study period both in light of chrysophyte-based reconstruction of a number of days below 4 °C (DB4 °C) representing the severity of winter and chironomid-based reconstruction of August temperatures (Fig. 10F,G). Particularly cold was the first half of the 11[th] century. This century was also colder in comparison to the contemporary period (1951–2000). For the 12[th] century, we have also data from old (Szychowska-Krąpiec, 2010, Fig. 56) and new (see Figs 9 and 10B,C) reconstructions of air temperature from Lesser Poland based on fir and pine chronologies. The 12[th] century was markedly warmer than the 11[th] century and also warmer than at present. In particular, the summer seasons in the first half of this century were very warm (August temperature about 18 °C, Fig. 10G), while in the second half they were close to the 1951–2000 norm (16.9 °C). As results from Fig. 10F,G, the 12[th] century was the warmest century in the entire study period in terms of summer temperature, while the cold-half year temperature was slightly colder (except the last 2–3 decades) than in the next three centuries. The reconstructions of winter temperature for Lesser Poland based on dendrochronological data (Fig. 10B,C) is characterised by strong fluctuations oscillating from about 0 °C to -2 °C relative to present means. But on average, winter conditions were warmer than in the 13[th], 14[th] and 15[th] centuries.

Proxies represented the entire cold season (see Fig. 10 D,F), revealing the existence of warm temperatures in the 13[th] century that were particularly high in its first half. The Scots pine reconstructions of late winter and early spring (Feb–Mar) both for Lesser Poland and the Kuyavian-Pomeranian region (Figs 9 and 10A–B,D) generally also confirm this finding. The average temperature of this season was only slightly colder (by 0.2 °C) than at present. Summer temperatures in the 13[th] century oscillated below and above the long-term contemporary norm. Similarly to winter temperature, summers were also on average slightly colder than at present by the same value. However, this century was significantly colder (by 0.3–0.6 °C) than the 12[th], 14[th] and 15[th] centuries.

Good agreement exists for the cold season temperature dendrochronological reconstructions for Poland in the 14[th] century. They indicate the existence of colder conditions in the 1330s and the last three decades of the century and warmer conditions in the middle of the century (see Fig. 10). On the other hand, values of the reconstructed number of DB4 °C are slightly lower than the norm for the period 1001–2000 (100.3 days), and significantly lower than the present norm (102.7). That is, winters were mild and thermally stable in this time (see Fig. 10F). Summer temperature was relatively high in almost the entire century and very high in the last decade, when it was comparable to that in the first half of the 12[th] century (see Fig. 10G). On average, the temperature was only slightly lower than in the 12[th] century (by 0.1 °C), but it was higher than in the latter half of the 20th century (by 0.4 °C).

For the 15[th] century, we have many excerpts describing the weather. Based on these sources, winter and summer temperatures were reconstructed (see Fig. 5). They document the existence in that century of very cold winters and very warm summers, which means that climate continentality was very high. Sadowski (1991) estimated that Poland's climate continentality greater in the 15[th] century than at any other time in the period 1201–1980 that he had analysed. The chironomid-based reconstruction of August temperatures (Fig. 10G) shows a similar time pattern as the summer temperature changes presented in Fig. 5. In both reconstructions, higher temperatures were observed in the second half of the century, and particularly in the 1470s. In the case of winter temperatures, all dendrochronological reconstructions generally confirm that they were usually lower than at present (Fig. 10), although large oscillations are seen. Some contradictions, however, are noted regarding the occurrence of the two coldest decades in the reconstruction based on documentary evidence (the 1430s and 1450s). The first cold decade was noted in Lesser Poland, but not in the northern and central Poland. In the latter areas, very warm winters occurred in this time. Similar results are shown by the reconstruction of the number of DB4 °C. At this time, they were fewest (only 96 days) of the entire study period (Fig. 10F) and even the entire millennium (Hernández-Almeida et al., 2015). On the other hand, there is correspondence of the thermal character of winter between the reconstruction based on documentary evidence and the number of DB4 °C for the second half of the 15[th] century. The coldest winters (in the 1450s) were significantly better registered in trees growing in northern and central Poland, while in southern Poland a warming was even noted (Fig. 10). The decrease in winter temperature in Poland in the 1430s and 1450s was confirmed by the reconstruction of the ice winter index in the Western Baltic Sea for the 15[th] century (Koslowski and Schmelzer, 2007).

We observed that, for Scots pine from Lesser Poland and northern Poland, negative pointer years outnumbered positive pointer years. This was probably caused by the influence of unfavourable climatic conditions prevailing in the study period, and long and severe winters. For the period 1361–1500, 38 pointer years were found for which information about weather conditions was available in the historical sources. For as many as 71% of pointer years, descriptions of extreme weather conditions were found that favoured the annual tree growth being either low (e.g., severe and long winter, drought) or high (e.g., warm winter, heavy rainfall).

The interpretation of climate based on pointer years in oaks is more difficult because they depend simultaneously on temperature and precipitation. Krąpiec (1998) found that positive pointer years for oaks in Poland are associated mainly with heavy rainfall in the growing season following a mild winter, while negative ones are linked to dry years with cold and snowless winters, and to spring frosts damaging the cambium. For example, the negative pointer years of AD 1314 and 1317 in Lesser Poland were caused by severe cold weather that caused a famine in Bohemia and neighbouring areas (Brázdil and Kotyza, 1997). The analysis of moon rings (MR) of oaks is easier to interpret, allowing long and severe winters to be determined. In the period 1361–1500, 19 such winters were recorded, and, for almost half (47.4%), the presence of moon rings was found.

As results from the above summary for Poland in medieval times, most of the proxies allow only for the temperature reconstruction of the entire cold half-year (Nov–Apr) or some of its sub-periods. Information about the weather in summer for the whole of the study period is limited to only one reconstruction of August temperature based on chironomid assembles (Fig. 10G). For late medieval times, some information is also available from documentary evidence. As a result, it is difficult to compare our results presented in the paper with those from the Northern Hemisphere and the European territory since proxy series of data (tree rings in particular) for the mentioned areas are biased towards the warmer seasons of the year (e.g., Table 1 in Ljungqvist, 2010). Fortunately, some comparisons are possible using existing reconstructions for European regions based on documentary evidence. However, due to the scarcity of available historical sources, the comparison should be limited mainly to late medieval times.

Comparison of August temperature from NE Poland (Fig. 10G) against summer temperature (Jun–Aug) reconstructed for Europe by Luterbacher et al. (2016) shows quite good correspondence. In both reconstructions, the coldest temperatures in the period 1001–1500 occurred in the 11 century and the mid-15[th] century. Cold summers were also noted from the mid-13[th] century to the mid-14[th] century. On the other hand, in both reconstructions, the markedly warmest summers occurred in the 12[th] century. However, in Poland they ended earlier than in Europe (in the middle part of this century) but started earlier, at the end of the 11[th] century. Smaller summer warmings in both areas were also noted at the turn of the 15[th] century and in the second part of the 15[th] century.

In Europe, however, the scale of warming in these times was significantly smaller than in the 12[th] century, while in Poland it was comparable across the periods, but the duration of the warming was shorter in both areas. The most comparison of the results is most reliable for the 15[th] century, for which we reconstructed summer temperature (Fig. 5). The second half of the century was clearly warmer than the first, not only in Poland but also in Czech Lands (Brázdil, 1994,

1996), Switzerland and in the entirety of Central Europe (Camenisch et al., 2016), as well as in Scandinavia (Gouirand et al., 2008). The decade 1471–1480 was the warmest in the entire 15[th] century in Poland (Fig. 5) and also in the Low Countries and Central Europe (Camenisch, 2015; Riemann et al., 2015; Camenisch et al., 2016), while the warmest summers occurred in the 1480s in Czech Lands (Brázdil, 1994, 1996) and in the 1490s in Switzerland (Trachsel et al., 2010; Camenisch et al., 2016). For the 15[th] century, there exists a correspondence between the run of the August temperature in Poland (Fig. 10G)

and reconstructions of summer temperature for some European areas based on documentary evidence and tree rings (Fig. 2; see also reconstructions 2–5 in Camenisch et al., 2016). Summer temperature was colder in the first part of the century than in the second.

Luterbacher et al. (2010) found that the correspondence of temperatures between Poland and Europe is lower for summer than for winter. In the latter season, there exists a very high correlation (0.96, at interannual and multidecadal time

scales) between reconstructed Polish and (non-Polish) European mean temperatures (e.g., excluding the grid points representing Poland) over the period 1500–2000. However, for Europe there are only some reconstructions based on documentary evidence (e.g., Brázdil, 1994, 1996; Pfister et al., 1996, 1998; van Engelen et al., 2001; Shabalova and Van Engelen, 2003; Glaser and Riemann, 2009; Camenisch, 2015; Camenisch et al., 2016; Retsö and Söderberg, 2019) and modeling works (e.g., Goosse et al., 2006; Schimanke et al., 2012). The scarcity of historical sources for the study period,

particularly before the 15[th] century (e.g., Brázdil, 1996; Glaser and Riemann, 2009; Oliński, 2022; see also Fig. 2), means that the reconstruction uncertainties can be significant. Nevertheless, a correlation exists for some periods between Poland's winter temperature and reconstructions for European areas based on documentary evidence.

For example, such agreement was observed in the 11[th] century, where also in western-central Europe, winters were lower than average (Pfister et al., 1998). Similar results were also found for Europe north of the Alps (Alexandre, 1987 after

Brázdil 1996) and Czech Lands (Brázdil, 1996), while the opposite was found in the Low Countries (van Engelen et al., 2001). In the 12[th] century, there is still good agreement with the reconstruction of the DB4, but less agreement with the reconstructions of Feb–Mar temperatures (compare Fig. 10 in Pfister et al. (1998) and Fig. 9 here). On the other hand, all reconstructions reveal a significant winter warming at the end of the 12[th] century that has also been noted in Germany (Glaser and Riemann, 2009) and the Low Countries (van Engelen et al., 2001). In the 13[th] century, changes in the

reconstructed temperature in Poland and west-central Europe differed much, particularly in the first half of the century. However, the warming of winters in this time occurred also in England, Ukraine and Russia, but not in Europe north of the Alps, Czech Lands (Fig. 4 in Brázdil, 1996), Germany (Glaser and Riemann, 2009) or the Baltic Sea Basin (Schimanke et al., 2012). On the other hand, a good coherence was usually stated for the 14[th] century, where in the two compared areas, colder conditions prevailed except in the middle of the century (compare Fig. 2 in Pfister et al. (1996)) with available

reconstructions from Poland, including new ones presented in Fig. 9. Also agreement is seen for Europe north of Alps, but not for Czech Lands, Ukraine or Russia (Brázdil, 1996) and the Low Countries (van Engelen et al., 2001). The reconstructed winter (DJF) temperature in Poland for the 15[th] century based on documentary evidence (Fig. 5) matches well with the data presented for Czech Lands, Russia, England (Brázdil, 1994, 1996), Baltic Sea Basin (Schimanke et al., 2012)., Sweden

(Retsö and Söderberg, 2019), the Low Countries (Camenisch, 2015), Alps (Mangini et al., 2005), and the whole of Europe
(Goosse et al., 2006), but only partly with data for Germany (Glaser and Riemann, 2009).

Ljungqvist (2010) found that the Northern Hemisphere proxy collection is biased toward the year's warmer seasons. The reason for this is that predominant among the proxies in the reconstruction process are tree rings taken mainly from mountain regions (see Pages2k Consortium 2013, 2017; Tardiff et al. 2019). Nevertheless, the reconstructed summer temperatures are frequently used to represent annual values (e.g., Cook et al., 2004; Moberg et al., 2005; Ljungqvist, 2010)
due to the high correlation between seasons on decadal and longer time scales within the instrumental period. However, for smaller areas that are dominated by lowlands – like Poland – this is not the case. There is lack of correlation between mean summer and winter temperatures for Poland in 1901–2000. The correlation of summer temperature with the annual temperature is 0.4 and is two times smaller than the correlation of winter temperature (0.79). Therefore in Poland, winter temperature represents the annual temperature better than does summer temperature. It is thus justified to use winter
temperatures as annual proxies and thus to use them to distinguish the occurrence or not of the MWP (MCA) in Poland.

Due to the coverage of the entire millennium and also of the whole cold season, the best proxy that can presently be used to delimit the MWP in Poland is DB4 (Fig. 10F). These data locate this period in the time period 1180–1440 (excluding the period 1260–1270, partly a consequence of a volcanic eruption in 1268), when the winters were generally shorter and warmer than today (1951–2000). The other proxies usually do not allow the beginning of the MWP to be distinguished (due
to the reconstructed data series starting too late) but only its end. There exist, however, some essential differences between proxy results. The MWP is also suggested to have ended in the first half of the 15[th] century (e.g., in line with DB4 data) by Sadowski (1991). Some other authors concluded that the MWP in Poland ended at the beginning of the 14[th] century (Maruszczak, 1991; Kotarba, 2004; Balanzategui et al., 2017; Szychowska-Krąpiec, 2010 only fir chronology) or even the beginning of the 13[th] century (Koprowski et al., 2012). Based on new February–March reconstructions (Fig. 9), it is possible
to distinguish the MWP (from the end of the 13[th] century to the mid-15[th] century) only in series representing the Kuyavia-Pomerania region. These are thus in correlation with DB4 data, except the first 100 years. All the mentioned quantitative reconstructions suggest that the MWP in Poland was comparable to or warmer than the current temperature (1951–2000). For Europe, various time frames are given for the MWP, but most suggest that this period ended in the 14[th] century (e.g., Brázdil, 1996 for Czech Lands; Glaser and Riemann, 2009 for Germany; Millet et al., 2009 for northern French Alps;
Niemann et al., 2012 for Swiss Alps; Niedźwiedź et al., 2015 for the Baltic Sea Basin). The simulation of mean winter European temperatures presented by Goose et al. (2006, their Fig. 3) locates the MWP as extending until the first half of the 15[th] century; however, when we take mean summer temperature, the MWP ended in the beginning of the 13[th] century. The lack of spatial coherence in timing of the MWP across the entire globe was also recently confirmed by Neukom et al. (2019) based on proxy data published by Pages2k Consortium (2017). They locate the MWP from 751 to 1350, and according to
their Fig. 3, within this time span the peak warming in Poland occurred between 800 and 1000. However, we need to add that the Pages2k databases contain not a single proxy data series taken from Poland.

The warm/cold periods observed in Poland during the study period are an effect of natural variability in the climatic system (see the course of the set of forcings presented in Fig. 1 in Goose et al. 2006). For example, the low temperatures in the 11th century can be connected with negative anomalies of total solar irradiance (TSI), while in the 15th century they can also be associated with increased volcanic activity and the negative phase of the North Atlantic Oscillation (NAO) (Trouet et al., 2009). For example, the very cold decade of the 1450s observed both in Europe (Goose et al., 2006) and globally (e.g., Tardif et al., 2019) is most often explained by the Kuwae eruption of 1452 or 1453 (Gao et al., 2006). According to those authors "the Kuwae eruption was the largest stratospheric sulfate event of that period [since 1300, authors], probably surpassing the total sulfate deposition of the Tambora eruption of 1815, which produced 59 kg $SO_4$/km$^2$ in Antarctica and 50 kg $SO_4$/km$^2$ in Greenland." The eruption probably changed atmospheric circulation in the North Atlantic leading to the occurrence of the negative phase of the NAO (see Fig. 2 in Trouet et al. 2009), which is important in shaping the climate in Poland, as we show in the next paragraph.

Any extreme thermal condition in Poland in a season or entire year is most often caused by anomalous behaviour of atmospheric circulation, and only infrequently by volcanic activity. According to investigations made by Przybylak et al. (2003) for the period 1500–1990, the main cause of cold winters in Poland in this time was the negative phase of the NAO (in which advection of cold air from the north and east dominates), while warm winters were observable when the NAO was in the positive phase (advection from the west dominates). So too, the occurrence of summers that are extremely warm or very warm in Poland is also connected with changes in atmospheric circulations. However, cold summers are caused by advection from the west and north, and warm summers with southerly and easterly advection (Bartoszek, 2017, see his Table 3). It is reasonable to assume that the relationships observed in the contemporary summer climate of Poland also obtained in the study period.

## 5. Conclusions and final remarks

The main results of the present paper can be summarised as follows:

1) Since the 1990s, significantly more quantitative reconstructions of climate in Poland have been published using different kind of proxies covering the study period, including four new ones presented in the paper.

2) The areally averaged summer temperature for Poland in 1901–2000 correlates with the annual temperature at r=0.4, which is smaller than the correlation with winter temperature (r=0.79). Therefore in Poland, winter temperature better represents the annual temperature than does summer temperature. It is justified to use winter temperature as annual proxies and thus to use them to distinguish the occurrence or not of the MWP (MCA) in Poland.

3) Analysis of all available reconstructions reveals the existence of the MWP in Poland from the late 12th century to the first halves of the 14th or 15th centuries. We found that the MWP in Poland was comparable to or warmer than the current temperature (1951–2000).

4) The coldest conditions in the entire study period were noted in the first half of the 11th century (both winter and summer) and the second half of the 15<sup>th</sup> century (only winter).

5) The best and most reliable knowledge about climate in the study period exists for the 15<sup>th</sup> century, for which we present winter and summer reconstructions based on documentary evidence (Fig. 5). All reconstructed 10-year mean values of winters and summers were, correspondingly, colder and warmer than today. The coldest winters occurred in the 1450s, 1430s and 1490s, while the warmest summers occurred mainly in the 1470s. Thus our reconstruction confirms the conclusion formulated earlier by Sadowski (1991) that the 15<sup>th</sup> century in Poland was characterised by the highest 525 climate continentality in the entire millennium.

6) For the study period, information about wetness of seasons in Poland is available only based on documentary evidence, and mainly for the 15<sup>th</sup> century. In line with expectations, the most such information is available for summer and then for winter, while it is only very rarely available for the transitional seasons (Fig. 4). In the 15<sup>th</sup> century, all categories of wet winters dominated, with a maximum in 1430s (4). This means that the 1430s were characterised in Poland by very 530 cold and snowy winters. Except for the last three decades of the 15<sup>th</sup> century, extremely wet and very wet conditions also dominated in summer, and especially in the 1460s (5 cases). Radical dryness of the air occurred in the 1470s, when five extremely dry or very dry summers were noted (Fig. 4).

7) Good agreement was found between the reconstructions of Poland's climate from 1001–1500 and many reconstructions available for Europe, which is in line with findings presented by Luterbacher et al. (2010) for the period 1500–2000.

The further improvement of the knowledge of the climate in Poland in the first half of the last millennium presented here will be closely connected with findings based on new proxies from the natural archives – mainly tree-ring wood density and stable isotopes and other biological proxies. Based on the preliminary archival research and the survey of the library holdings, we have conducted in the last 20 years, we are certain that it is impossible to find a significant amount of further information about weather in historical sources, particularly prior to the 15<sup>th</sup> century.

540

**Author's contribution.** Conceptualization: RPr, PO, MKo; Methodology: RPr, PO, MKo, MKr, ESK; Validation: RPr, PO, MKo, MKr, ESK; Formal analysis: RPr, PO, MKo, MKr; Investigation: RPr, PO, MKo, MKr, ESK, RPu, AP; Resources: RPr, PO, MKo, MKr, ESK, AP; Data Curation: RPr, PO, MKo, MKr, ESK, AP; Writing - Original Draft: RPr, PO, MKo, MKr, ESK; Writing - Review & Editing: RPr, PO, MKo, MKr, ESK, RPu, AP; Visualization: RPr, PO, MKo, MKr, RPu; 545 Project administration: RPr; Funding acquisition: RPr, ESK and MKr.

**Competing interests.** The contact author has declared that none of the authors has any competing interests.

**Financial support.** The research work of PO and RP was supported by grants funded by the National Science Centre, Poland (Grants Nos 2013/11/b/HS3/01458, 2020/37/B/ST10/00710). MK and RPu were supported by a grant funded by the National Science Centre, Poland (Grant No 2020/37/B/ST10/00710). ESK and MKr were supported by the grant of the Faculty of Geology, Geophysics and Environmental Protection of AGH University of Science and Technology in Cracow no. 16.16.140.315.

**Data Availability.** The tree-ring chronologies supporting the conclusion of this study are available from ESK, MK and MKr upon request.

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

Table 1. Statistics of chronology and climate reconstructions.

| Chronology and its time span | Number of samples | eps | snr | Reconstructed parameter | Calibration period | Calibration statistics | Verification period | Verification statistics | | | Model for whole period |
|---|---|---|---|---|---|---|---|---|---|---|---|
| Scots pine. Kuyavia-Pomerania; 1168–2015 | 285 | | 28.76 | Feb–Mar temperature | 1871–1943 | r=0.477, p<0.05 | 1944–2015 | RE= 0.159 | CE= 0.133 | prediction RMSE= 0.153 | 0.45, p<0.05 |
| Scots pine. Lesser Poland; 1091–2011 | 285 | 0.97 | 28.70 | Feb–Mar temperature | 1846–1960 | r=0.44, p<0.05 | 1961–2000 | 0.134 | 0.116 | 0.384 | 0.39, p<0.05 |
| Silver fir. Lesser Poland; 1109–2017 | 484 | 0.98 | 61.34 | Dec–Mar temperature | 1846–1960 | r=0.57, p<0.05 | 1961–2000 | 0.036 | 0.028 | 0.312 | 0.49, p<0.05 |

Table 2. Frequency of occurrence of extremely warm and wet, as well as cold and dry, winters (DJF) and summers (JJA) in Poland from 1361 to 1500.

| | Air temperature | | | | Atmospheric precipitation | | | | Extreme situations | |
|---|---|---|---|---|---|---|---|---|---|---|
| | DJF | | JJA | | DJF | | JJA | | | |
| Period | 2 & 3 | -2 & -3 | 2 & 3 | -2 & -3 | 2 & 3 | -2 & -3 | 2 & 3 | -2 & -3 | Total | % |
| 1361–1400 | 0 | 4 | 1 | 1 | 2 | 0 | 1 | 1 | 10 | 7.1 |
| 1401–50 | 5 | 18 | 4 | 4 | 7 | 3 | 8 | 4 | 53 | 37.6 |
| 1451–1500 | 5 | 19 | 16 | 5 | 7 | 3 | 10 | 13 | 78 | 55.3 |
| 1361–1500 | 10 | 41 | 21 | 10 | 16 | 6 | 19 | 18 | 141 | |
| % | 7.1 | 29.1 | 14.9 | 7.1 | 11.3 | 4.2 | 13.5 | 12.8 | | 100 |

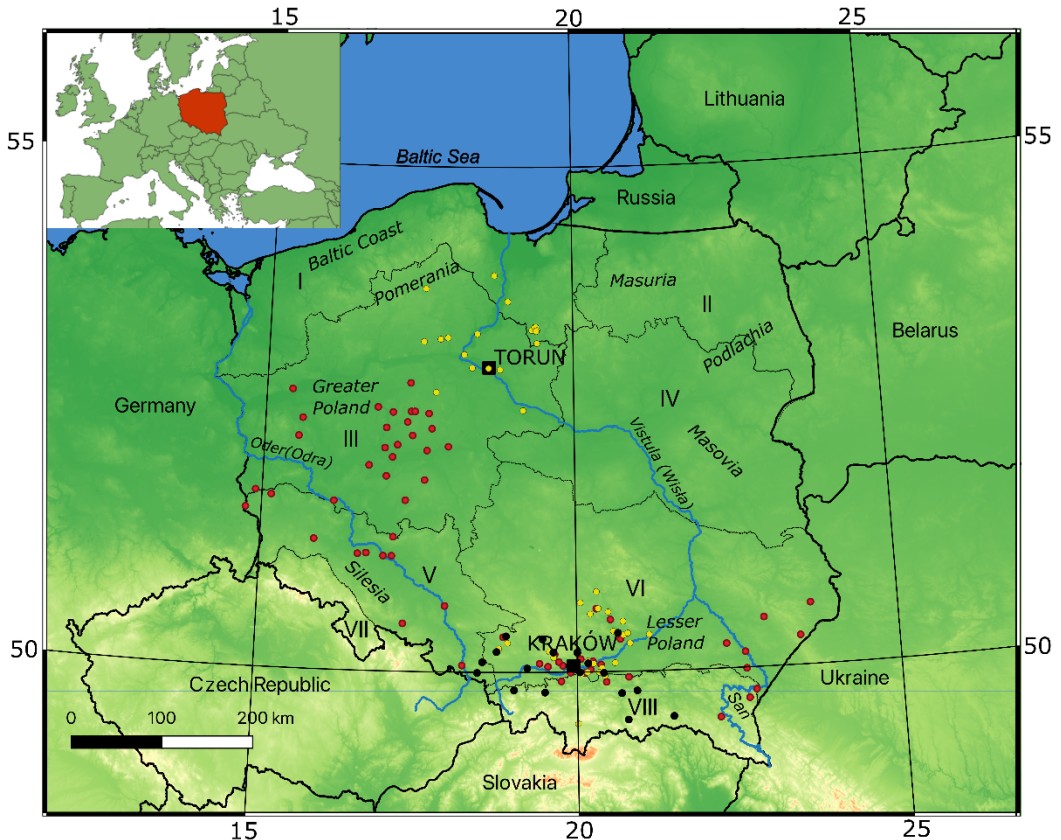

**Figure 1.** Study area with historical and natural-forest regionalisation and location of materials used for research. Natural-Forest Provinces (Zielony and Kliczkowska 2012): I – Baltic Coast province II – Masuria–Podlachia province, III – Greater Poland–Pomerania province, IV – Masovia–Podlachia province, V – Silesia province, VI – Lesser Poland province, VII – Sudetia province, VIII – Carpathia province. Dendrochronological sites: yellow dots – pine, red dots – oak, black dots – fir. Black squares – meteorological stations

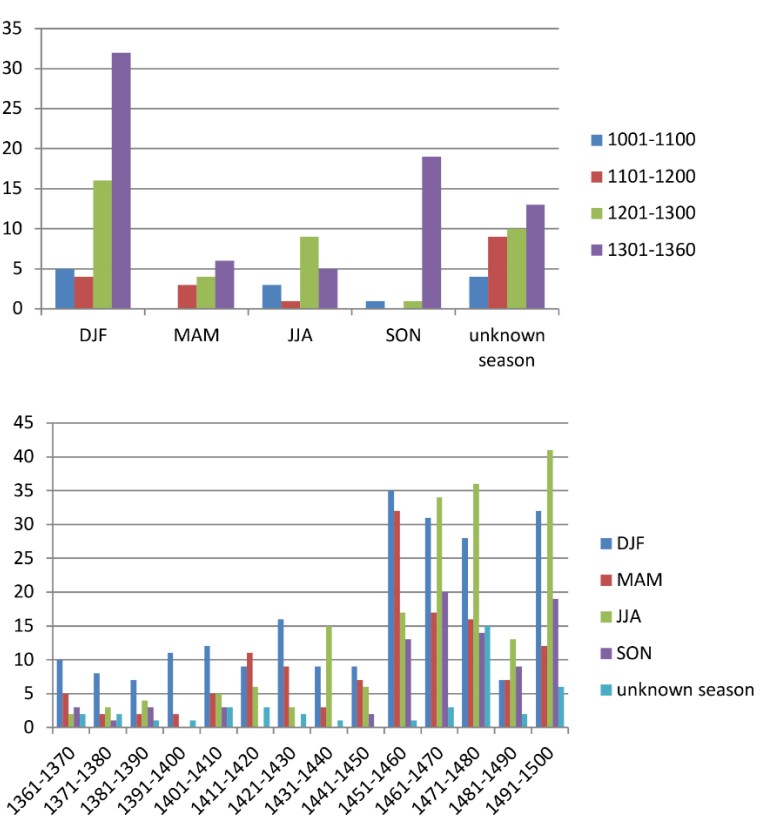

**Figure 2.** Number of weather notes/excerpts for Poland available in historical sources in the periods: *above*, AD 1001–1360; *below*, AD 1361–1500

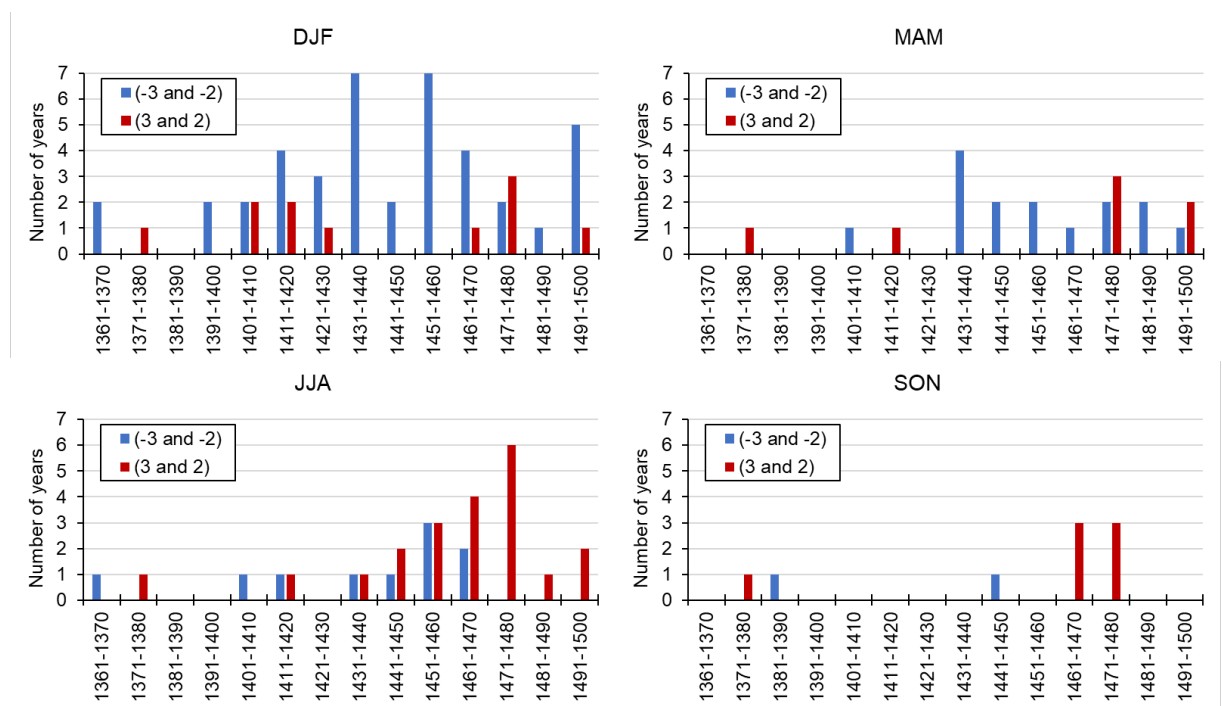

**Figure 3.** Decadal frequencies of occurrence of winters (DJF), springs (MAM), summers (JJA) and autumns (SON) that were extremely cold and very cold (indices -3 and -2) and extremely warm and very warm (indices 3 and 2) in Poland between 1361 and 1500

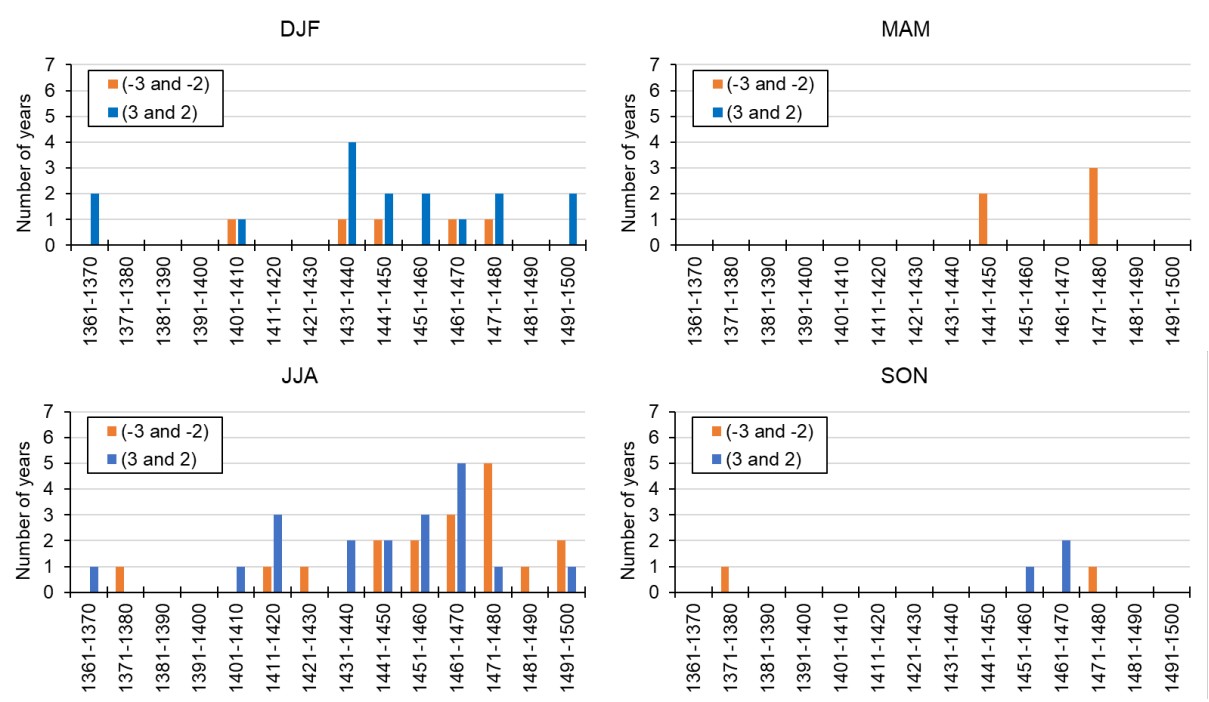

**Figure 4.** Decadal frequencies of occurrence of winters (DJF), springs (MAM), summers (JJA) and autumns (SON) that were extremely dry and very dry (indices -3 and -2) and extremely wet and very wet (indices 3 and 2) in Poland between 1361 and 1500

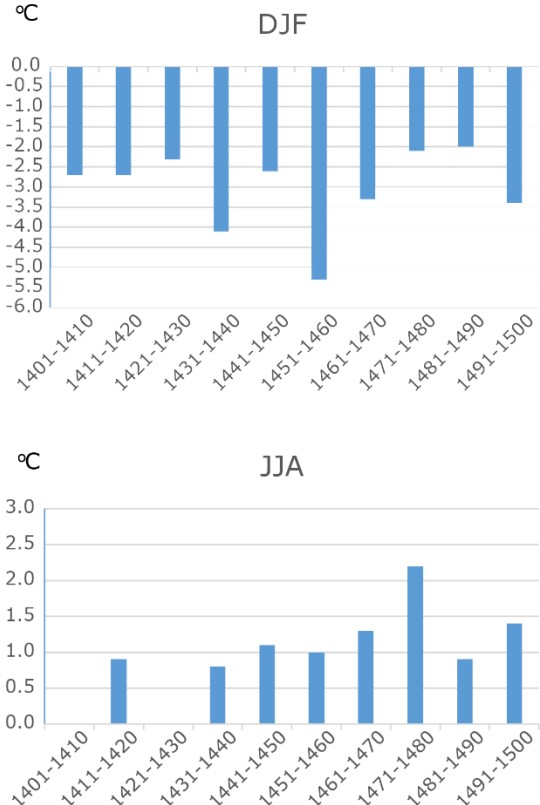

**Figure 5.** Anomalies of 10-year mean values of winter (DJF) and summer (JJA) air temperature in Poland in 15th century relative to means from 1951–2000 reference period. For reference period. Data were taken after Kożuchowski and Żmudzka (2003).

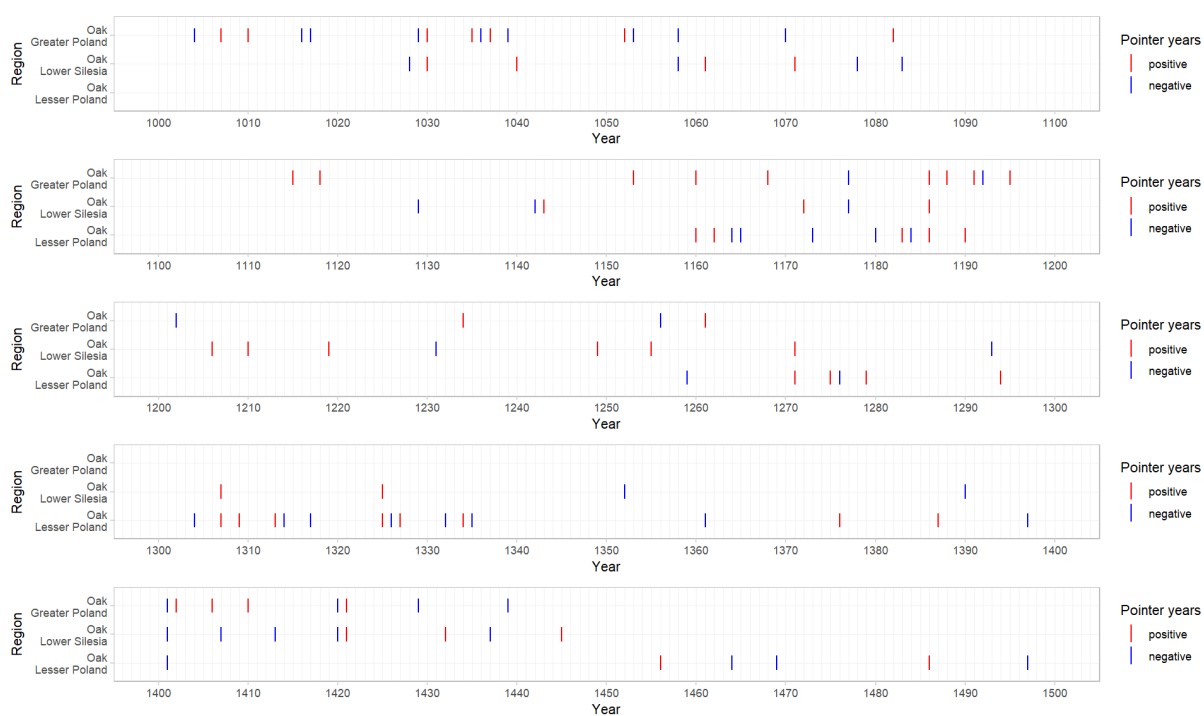

**Figure 6.** Pointer years in oak growing in Greater Poland, Lower Silesia and Lesser Poland, 1001–1500

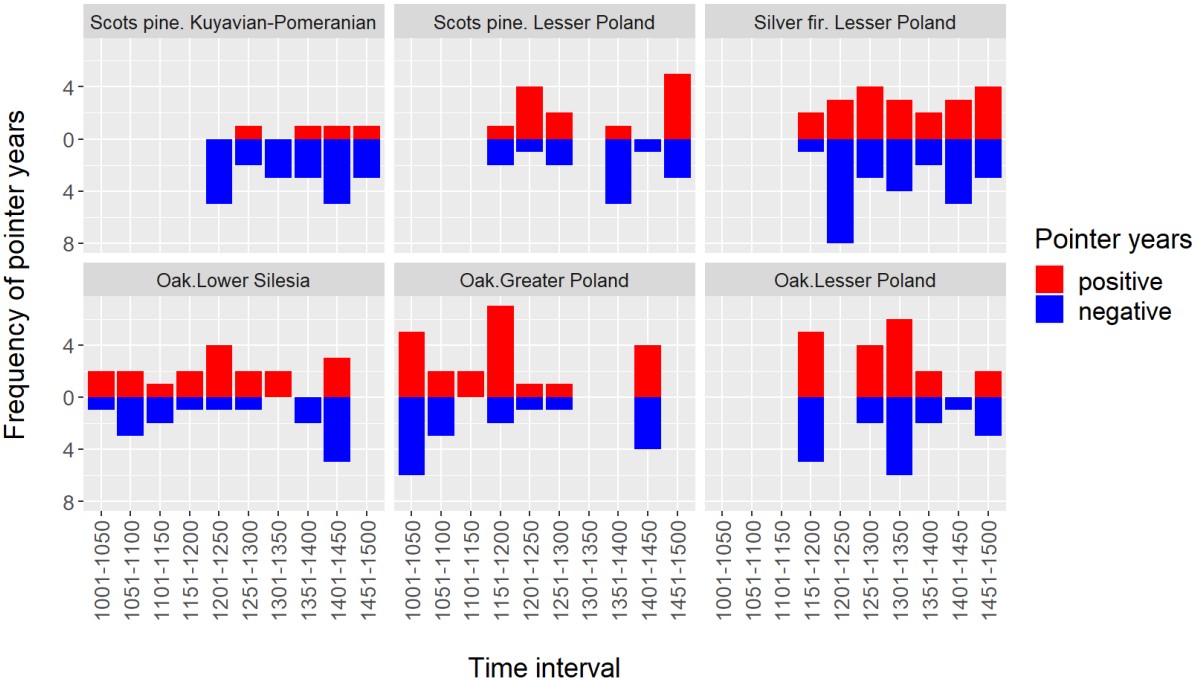

**Figure 7.** Frequency of occurrence of pointer years in trees (Scots pine, silver fir and oak) in 50-year periods in Poland, 1001–1500

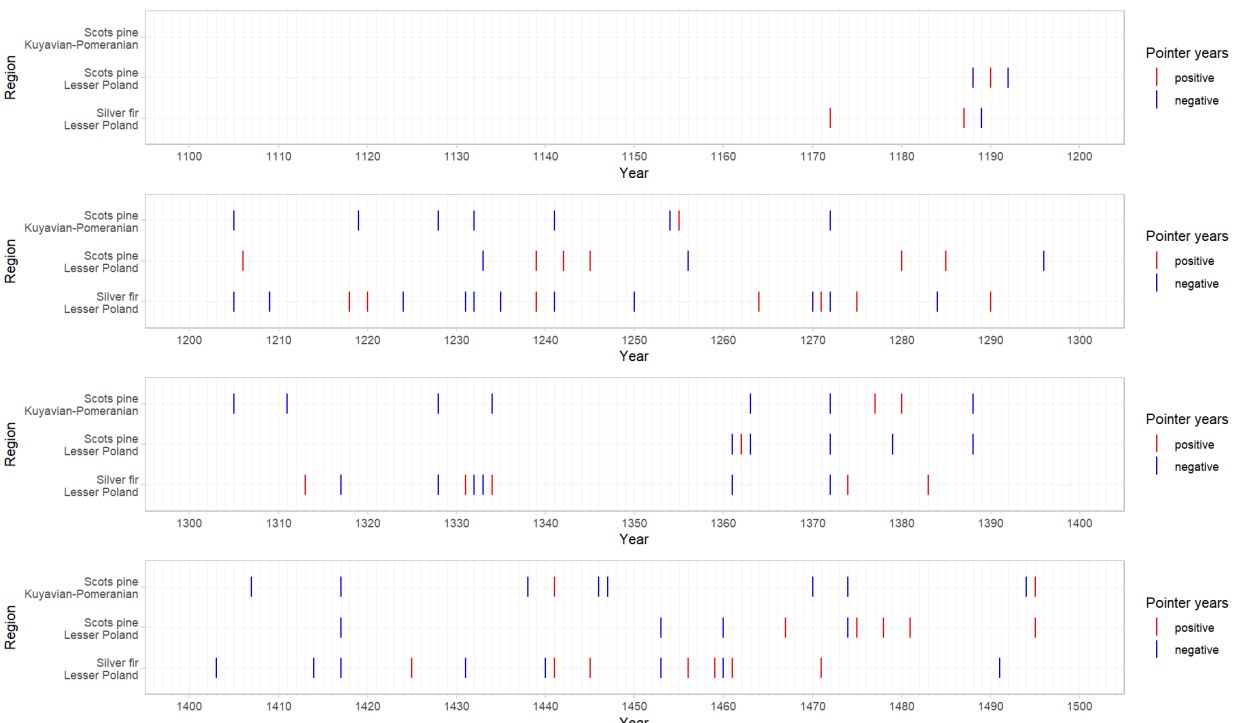

**Figure 8.** Pointer years in trees (Scots pine and silver fir) growing in Kuyavia-Pomerania and Lesser Poland, 1101–1500


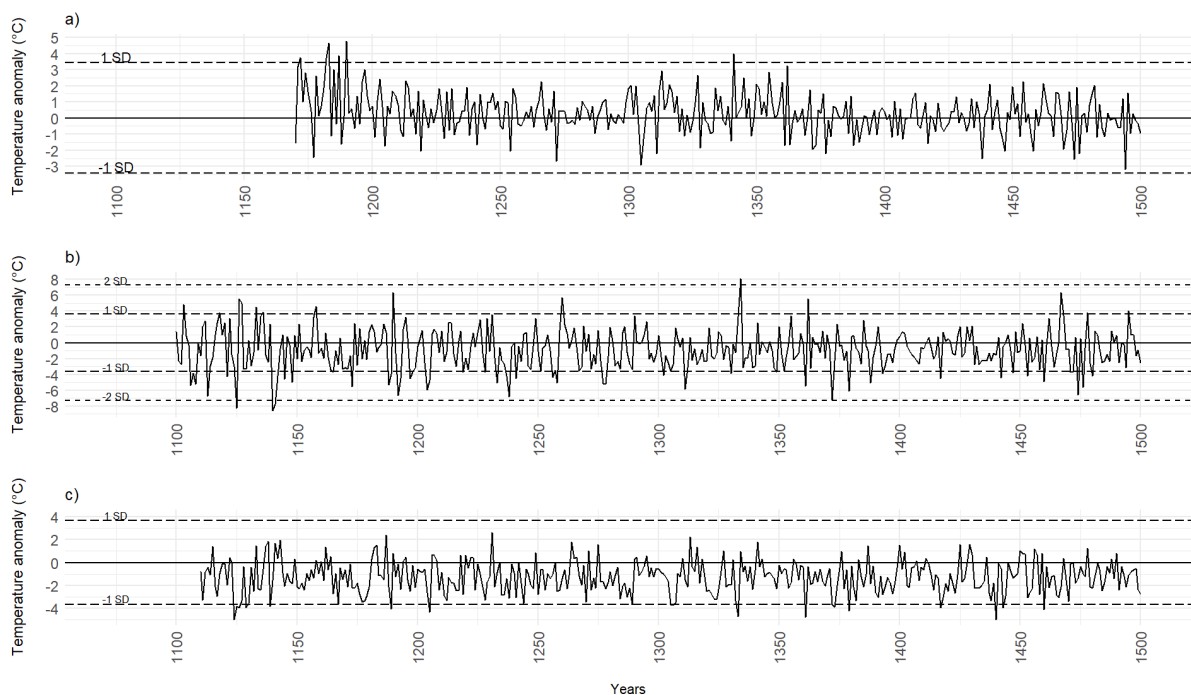

**Figure 9.** Reconstruction of average air temperature (°C): for February–March in: (a) northern Poland and (b) southern Poland, based on the width of pine rings; and (c) for December–March in southern Poland based on fir-ring widths. Anomalies were calculated relative to the period 1951–2000. Standard deviations were calculated using data from 1951–2000


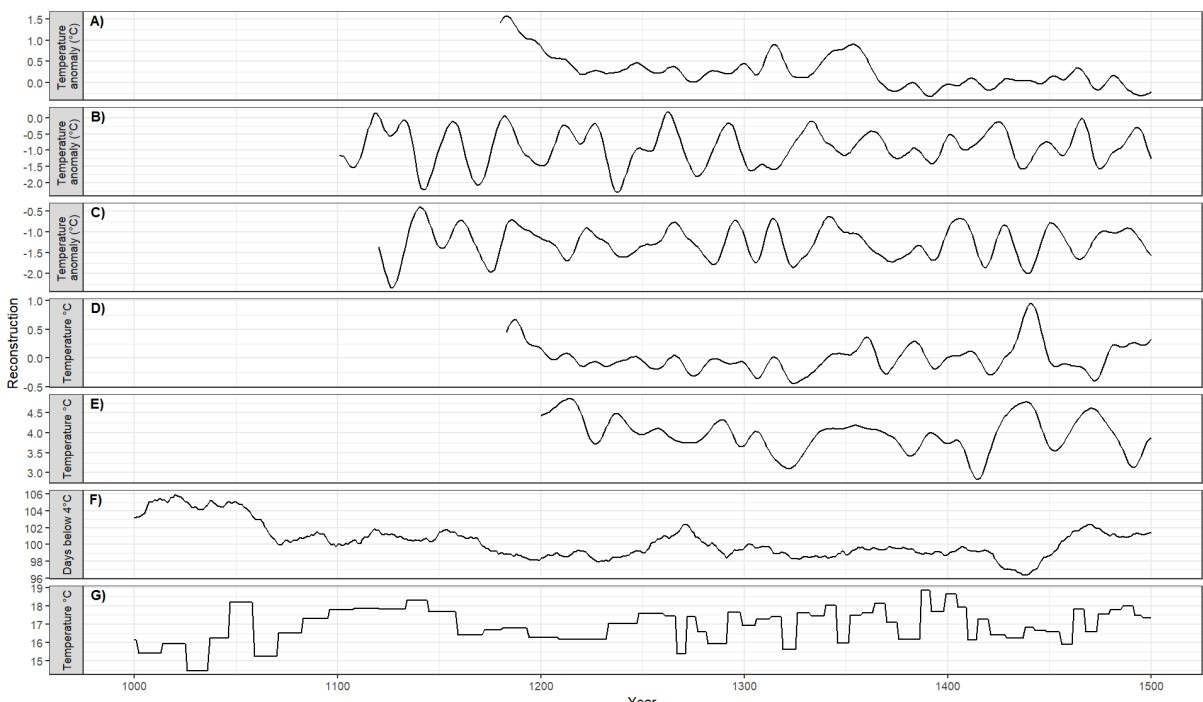

**Figure 10.** Reconstruction of air temperature in Poland in moving windows: A) Feb–Mar temperature anomaly in a Gaussian 20-year moving window for Kuyavia-Pomerania region. B) Feb–Mar temperature anomaly in a Gaussian 20-year moving window for the Lesser Poland region. C) Dec–Mar temperature anomaly in a Gaussian 20-year moving window for the Lesser Poland region. D) Feb–Mar temperature anomaly in a Gaussian 20-year moving window for Kuyavia-Pomerania (modified after Koprowski et al. 2012). E) Nov–Apr temperature anomaly in a Gaussian 20-year moving window for northern Poland (modified after Balanzategui et al. 2017). F) chrysophyte-based reconstruction of number of days below 4 °C (Hernańdez-Almeida et al. 2015). G) chironomid-based reconstruction of August temperature (Hernańdez-Almeida et al. 2017)