# Peer review of "The climate in Poland (Central Europe) in the first half of the last millennium, revisited"

_EGUsphere, 2023_

## Author Comment (AC1)

Journal: Climate of the Past

Manuscript ID: egusphere-2023-1143

Title: The climate in Poland (Central Europe) in the first half of the last millennium, revisited

Dear Professor Natalia Piotrowska

Editor, Climate of the Past

We would like to thank you and the anonymous reviewers for providing positive feedback and constructive comments on our manuscript. All comments were carefully considered, and we believe they helped us improve the description of our work. The detailed corrections/modifications are listed below, point by point.

(Note: The changes in the text and the answers to the reviewer's questions/suggestions are marked in red font. We revised the text taking into account all comments and suggestions proposed by the reviewers. All changes have been carefully applied to the text.)

**Response to Reviewer No. 2**

Review of manuscript "The climate in Poland (Central Europe) in the first half of the last millennium, revisited".

This manuscript investigates climate variability in Poland over the first half of the last millennium, especially over the 15th century. The authors combine previously published climate reconstructions with novel time series from dendrochronological and documentary material. I am especially impressed with the extensive and detailed work on gathering all the documentary evidence (Tables S2 and S3).

ANS: Thank you.

As I do not have expertise in dendrosciences, I will focus mostly on the documentary evidence and the reconstruction created from these data. After taking into consideration the few suggestions pointed out below, I think this manuscript is well suited to EGUsphere.

Major

My major concern is related to the temperature reconstruction based on documentary evidence (lines 112–116 and 235–249, Fig 4.). The authors state that the reconstruction method is "described in a paper by Przybylak et al. (2005) and is therefore omitted here". However, I would strongly suggest including a brief description on the method, as well as adding some critical reflections on the potential biases of historical data. There are few reasons for this:

ANS: Thank you for these suggestions. We added the following passage:

*Standard methods used commonly in the historical climatology were utilised for this purpose. Pfister et al. (1994) proposed that, for the following index values: i) +3 and -3 should represent anomalies exceeding 2.0 standard deviations (SD) from the mean of the long-term period, ii) +2/-2 and +1/-1 should represent less extreme conditions, i.e. 1.41–2.00SD and 0.7–1.4SD, respectively, iii) 0 (>-0.7SD – <0.7SD) should represent the average climate of the long-term period or missing data. The above-mentioned criteria were slightly modified by us using results of calibration of documentary evidence with air temperature data from Warsaw for the period 1789–1850 done by Sadowski (1991). As a result, the seasons described in this paper by indices from +3 to -3 fulfil the following criteria:*

$$+3 \geq m + 1.5SD$$

$$m + 1.0SD \leq +2 < m + 1.5SD$$

$$m + 0.5SD \leq +1 < m + 1.0SD$$

$$m - 0.5SD < \ \ 0 < m + 0.5SD$$

$$m - 1.0SD < -1 \leq m - 0.5SD$$

$$m - 1.5SD < -2 \leq m - 1.0SD$$

$$-3 \leq m - 1.5SD$$

*where m is the long-term mean (for calibration period 1789–1850) air temperature from the Warsaw series and SD is the standard deviation of that series. For more details, see Przybylak et al. (2005). The reader should be reminded of the possible biases related to climate reconstruction based on the documentary evidence such as, for example: i) the number of available sources, their quality and discontinuous structure, ii) the subjectivity of the indexation, iii) weaknesses of the reconstruction method used. A detailed overview of the strong points and the drawbacks of documentary data is presented by Brázdil et al. (2005).*

We hope that the reconstruction method and documentary data are now more clearly presented.

1) As the documentary evidence presented by Przybylak et al. (2005) covered years 1501–1840, the transformation from the -3,...+3 index values into °C (Przybylak et al. 2005, Tables II and III) was done with a reference period when the documentary and meteorological data overlapped. This is not the case in the current study, as (as far as I understood) reference period 1951–2000 is used for the 1361–1500 reconstruction. Thus, is the method still applicable if the documentary and meteorological series do not overlap?

ANS: In the present paper, as well as in our previous reconstructions published in papers Przybylak et al. (2005) and Przybylak (2011), the same methodology was used. That is why our reconstructions for the entire historical period (including the present reconstruction for the 15th century) are fully homogenised and comparable. Calibration and verification of documentary evidence with meteorological data from Warsaw from the period 1789–1850 was made by Sadowski (1991). Only for this period is documentary evidence available that overlaps with the meteorological data. In the next step, for each index we calculated mean values of seasonal temperatures using the homogenised

temperature series from Warsaw from the period 1779–1999 and using established temperature intervals as shown above to the first point of question, i.e. also covering the reference period 1951–2000 (excluding one year). Nobody is trying to make a calibration and verification procedures based on contemporary documentary evidence and also contemporary meteorological data, which if we understood correctly, the reviewer is proposing. Brázdil et al. (2005) clearly write: "Each kind of documentary proxy data needs to be calibrated against (early) instrumental series and is restricted to specific periods of the year in the same way as natural proxy data."

In the paper Przybylak et al. (2005), we calculated temperature anomalies between historical periods in reference to the two periods 1789–1850 (early instrumental) and 1901–60. Here, we decided to present the comparison to only more contemporary thermal conditions in Poland (1951–2000) taking into account areally average temperature for Poland calculated based on data from 45–50 weather stations. This reference period was used only to show the change in temperature between historical periods and the contemporary period.

2) The approach described by Przybylak et al. (2005) is based on the assumption that "there were no significant changes in mean temperatures […] from the 16th to the 19th centuries" and thus "it can be assumed that any changes in temperature variance that occurred were insignificant" (Przybylak et al. 2005, 778). However, can we similarly assume that there were no changes in the mean and/or variance between the periods 1361–1500 and 1951–2000?

Ans: True, we cannot make such an assumption. But we do not need to make this assumption in order to reconstruct temperature for the 15th century. The period 1951–2000 is used only for comparison of reconstructed mean decadal temperature from historical period against the contemporary observed values, in order to show the change in temperature between the two periods.

3) The reconstruction (Fig. 5) indicates that 15th century winters were systemically colder and summers warmer than during the later half of the 20th century, which provides further evidence on the transformation to a more continental climate during this period (lines 326–329). However, I was wondering whether the source material might contribute to the higher occurrence of colder/hotter anomalies as well? It is well established that historical weather descriptions contains bias in the focus of the observer and they emphasise extreme events (see, e.g., Brázdil et al. 2010, "European climate of the past 500 years" Climatic Change 101). The number of available weather descriptions (Fig. 2) and the frequency of extreme events (Table 2) show a similar temporal pattern. Thus, to what degree the increased number of mentioned cold/hot seasons can be explained by increasing source availability, especially because we know that historical records rarely record "average" weather?

Ans: The remark is important. Please note that the pattern of temperature changes between the 15th century and contemporary period is the same as in our reconstructions made for the period 1501–1840 (Fig. 3 in Przybylak et al. 2005). It supports the correctness of our recent reconstruction. You are right, evidently there exists some influence of a different number of weather notes on each reconstruction constructed based on documentary evidence. However, we do not know the scale of that influence. But we know that this influence is greater for the less-extreme events than for highly extreme events. The latter events were usually not missed in the historical sources even if the overall number of them was small (for the reason that you also mentioned). A very good example illustrating this is the analysis of the decades of the 1430s and 1450s. The number of weather notes for winter was about four times smaller in the first decade than in the decade 1450s (Fig. 2), but in both decades the number of extremely cold and very cold winters (indices -3 and -2) was the same (7 cases) (see Fig. 3). As a result, the reconstructed temperatures were also comparable (Fig. 4). Both decades were very

cold, and there is a lot of evidence for the occurrence of both cold periods in many reconstructions based on different proxies, even if averaged for the northern hemisphere or the globe.

4) And last, related to the point above, it would be good to add a table of criteria (i.e. list of typical descriptions/events in the documentary sources) for the -3,…,+3 classification for both seasons. This would be informative for readers who are not familiar with what is considered as "typical" or "extreme" weather over the study area. Moreover, clearly defined classification criteria would help to assess the high frequency of the late 15th century extreme events as well (Table 2).

ANS: Thank you for this suggestion. A new Table (Table S4) was added.

Moderate/minor

Please, provide further information on the meteorological data used, as the referred publication (Kożuchowski and Żmudzka, 2003) is not publicly available. Based on Fig 1, the meteorological data is coming from two stations, from Torun and Kraków?

ANS: The meteorological data from Toruń and Kraków were used only for the calibration and verification procedures needed for the reconstructions of temperature based on dendrochronological data. Kozuchowski and Żmudzka (2003) and Tomczyk (2022) used 45–50 and 40 weather stations to calculate areally averaged temperature for Poland, respectively.

The following sentence was added to the main text for clarity:

*These two present mean temperatures for the area of Poland were calculated based on data taken from 45–50 and 40 weather stations, respectively.*

Pay attention to terminology "current", "contemporary", and "present" (for example, lines 27, 239, 248, and 324). In many cases more accurate expression would be "latter half of the 20th century" or similar.

ANS: Thank you. The suggestion was introduced to the text.

The manuscript includes quite a lot of additional material from previous publications (e.g., Fig. 10 and the section Summary and discussion. Also the abstract starts by mentioning these materials). Consider introducing these data, for example, in a supplementary table.

ANS: Actually, only Fig. 10 includes data from previous publications. All other documentation is new and is shown for the first time. Figure 10 summarises concisely the course of temperature in Poland in the period 1001–1500, gathering all the existed temperature reconstructions for Poland published since 2010. For this reason, we believe this figure to be very important: it allows direct comparison of the results. Therefore, for the reader's sake, we prefer that the figure be left in the main body of the paper.

Technical

Line 22: Correct "date" as "data"

ANS: Done

Lines 66 and 67: Bold font for the titles.

ANS: Done

Table 1: Verification statistics are missing for Lesser Poland Scots pine and Silver fir.

ANS: Thank you. The verification statistics was added.

Line 149: What are the "extreme thermal conditions" that the moon rings indicate? Extremely cold winters? Please clarify for non-specialist.

ANS: Done

Lines 204 and 206. Winters 1280, 1306, and 1225. For example, in the case of winter 1280, does this mean the winter 1279/80 or 1280/81?

ANS: Done

Line 255 and after. What the identified moon rings indicate? Colder winters?

ANS: Done

---

## Author Comment (AC2)

Journal: Climate of the Past

Manuscript ID: egusphere-2023-1143

Title: The climate in Poland (Central Europe) in the first half of the last millennium, revisited

Dear Professor Natalia Piotrowska

Editor, Climate of the Past

We would like to thank you and the anonymous reviewers for providing positive feedback and constructive comments on our manuscript. All comments were carefully considered, and we believe they helped us improve the description of our work. The detailed corrections/modifications are listed below, point by point.

(Note: The changes in the text and the answers to the reviewer's questions/suggestions are marked in red font. We revised the text taking into account all comments and suggestions proposed by the reviewers. All changes have been carefully applied to the text.)

**Response to Reviewer No. 1**

Dear authors of the manuscript "The climate in Poland (Central Europe) in the first half of the last millennium, revisited", This study significantly furthers our understanding of climate variability in Poland during the first half of the last millennium. The new documentary records and the three tree-ring chronologies are noteworthy contributions to the paleoclimate research community. Archiving these records in a public database would substantially enhance the impact of this manuscript. However, the manuscript lacks a clear scientific question and a discussion of the related climate mechanisms. Here are my revised comments:

ANS: Thank you.

Main Comments:

The manuscript could be improved by introducing a clear scientific question or objective that it seeks to answer.

ANS: Done

It would also benefit from a detailed discussion on climate mechanisms related to the extreme events and warm periods. Please clarify how your current reconstructions differ from and contribute uniquely to the existing body of work. While reconstructions are valuable, it is essential to interpret and provide insights from these reconstructions, such as information about extreme events, rather than merely listing them.

ANS: All these suggestions were taken into account. Please see the text.

A detailed comparison, highlighting the similarities and differences between the tree-ring records and documentary records, could reinforce the benefits of using multiple proxies.

ANS: Such a comparison is already presented in detail in the *Summary and discussion* section. See text.

Specific Comments:

Lines 18-19: Clearly define the motivation and scientific question of your study.

ANS: Done.

Line 19: Consider incorporating all available quantitative climate reconstructions into your study, such as the gridded reconstructions (Neukom et al., 2019) and the reanalysis reconstruction (Tardif et al., 2019).

ANS: Thank you for this suggestion. Both items were included in the discussion part.

Line 50: The claim "Only a few papers also deal with a pre-1500 period" seems unsupported. Perhaps you could mention that there are significantly fewer studies dealing with the pre-1500 period compared to the post-1500 period.

ANS: You are right. We changed the text according to your suggestion. It should now should be clear.

Line 96: Please translate the information in Table S3 into English to facilitate the review process.

ANS: Done.

Line 183: Could you explain the reason for separating the sections before and after 1360 CE?

ANS: For pre-1361 we have significantly fewer sources, not enough to present the frequency of extreme categories of weather (3, 2, -2, -3), such as are presented in Figs 3 and 4. The following sentence was added for clarity: *It should be emphasized that there was a significant increase in the number of available historical written references from the 1360s onwards, and therefore 1360 was chosen as the threshold year for delimiting the two subperiods.*

Lines 209-210: If this sentence does not contribute significant information, consider removing it.

ANS: Done.

Lines 257-259: Please clarify the logic in this sentence.

ANS. Done. The sentence was corrected to: *The time distribution exhibits a few MRs coinciding across a larger territory simultaneously. These events are limited to the 15$^{th}$ century, when they were found in southern Poland (Wrocław, Kraków) in the 1440s and 1450s, and in Wrocław and Kutno in the early 1480s and 1490s.*

Lines 275-283: Please explain why tree-ring records respond to winter temperature in Poland, given that most tree-ring chronologies mainly respond to growing-season temperature, which directly impacts photosynthesis and the formation of growth rings (Fritts, 1976).

ANS: Please note that most dendrochronologies available for the world come from mountains or subpolar areas (see PAGES 2k Consortium: A global multiproxy database for temperature reconstructions of the Common Era, Sci. Data, 4, 170088, https://doi.org/10.1038/sdata.2017.88, 2017). For these areas, summer temperature influences the rate of tree growth, but our

dendrochronologies come from lowlands or uplands and, here, late winter and early spring temperature is critical (see Table 1, and more examples in Zielski et al. 2010, their Table 7.1).

The following passage was added to the text for clarity: *We confirmed (Table 1) the results presented earlier by Zielski (1997), who reveals statistically significant correlations between annual tree-ring widths (Kuyavia-Pomerania dendrochronology, 1891–1991) and the monthly mean air temperatures from the region, particularly from February and March, but also from January and April. Their values were equal to 0.47, 0.55, 0.26 and 0.18, respectively. This means that, in Poland, the low temperature occurring at the end of winter and at the beginning of spring has a strong negative influence on the width of tree-rings. On the other hand, precipitation has a weaker influence than temperature, though only in June and July is this statistically significant. More information about climate signal in selected trees in Poland is provide in Table 7.1 in Zielski et al. (2010).* Zielski A., Krąpiec M. Koprowski M., 2010, Dendrochronological Data, in Przybylak R, Majorowicz J, Brázdil R, Kejna M (eds). The Polish Climate in the European Context: An Historical Overview, Springer, Berlin Heidelberg New York, 191-217.

Lines 279-281: Please clarify what you mean by "the record today." Is it the instrumental record?

ANS: Thank you for this suggestion. Today means here period 1951-2000. We added this information to the text.

Lines 301-302: Consider including the instrumental temperature variability from 1951-2000 in Figure 10 A, B, C for a clearer comparison.

ANS: The series finishes at 1500 in Fig. 10, so we cannot add the period 1951–2000. This period was used by us to calculate temperature anomalies for the study period 1001–1500.

Lines 359-362: Comparing tree-ring chronologies directly could provide more insightful results.

ANS: Thank you for this suggestion. However, tree-ring reconstructions for Poland allow reconstruction of late-winter/early-spring temperatures (see Zielski 1997, Krąpiec 1998, Zielski and Krąpiec 2004, Przybylak et al. 2005 and Szychowska-Krąpiec 2010), while reconstructions available for other parts of Europe and the world most often reconstruct the summer temperature. Therefore, in our opinion, such comparison is not appropriate, because seasons often respond differently to climate change. See also Luterbacher et al.'s findings (lines 383-386 in the preprint) or Goose et al. (2006).

Lines 391-392: Please explain why the reconstruction matches the simulation.

ANS: Thank you for this suggestion. We have rethought this passage of the text and decided to delete this sentence. See the text.

Lines 438-439: This information does not seem to fit in the results section. Consider moving it to a more appropriate section.

ANS: Lines 438-439 are in the *Conclusion and final remarks' subsection* and not in the result section as the reviewer writes. Therefore we did not introduce any change.

Lines 440-430: There appears to be a logical inconsistency in this section. The relationship between seasonal temperatures does not seem to justify a replacement.

ANS: The lines mentioned by the Reviewer appear not to agree with the article as it is displayed to us, and we think that the reviewer is perhaps remarking on lines 440-443. If so, we do not share the Reviewer's opinion, and we still strongly maintain our view that the mean winter temperature in Poland significantly better represents the annual mean temperature than does the mean summer temperature. This is well understood by Polish climatologists. However, such relations are seen not

only in Poland but in all areas at polar and moderate latitudes, because the variability of winter temperature is significantly greater here than that of summer temperature.

Lines 462-463: This does not seem to be a conclusion.

ANS: Thank you for this suggestion. We changed little the sentence slightly to: *Good agreement was found between the reconstructions of Poland's climate from 1001–1500 and many reconstructions available for Europe, which is in line with findings presented by Luterbacher et al. (2010) for the period 1500–2000*. We hope that this sentence can now be treated as a conclusion.

The correlation coefficients in Table 1 for the tree-ring reconstructions are rather small, indicating that the explained variance in two out of three chronologies is less than 25%.

ANS: Please note that we used RE, CE and RMSE values for verification periods. In this case, CE and RE reach positive values and RMSE is below 1. The interpretation of these results was based upon previous research in life sciences.

McCarroll, D. Simple Statistical Tests for Geography; Chapman and Hall/CRC: Boca Raton, FL, USA, 2016; ISBN 9781498758819.

Cook, R.D. Using dimension-reduction subspaces to identify important inputs in models of physical systems. Proc. Sect. Phys. Eng. Sci. 1994, 9, 18–25.

Wilson, R.; Tudhope, A.; Brohan, P.; Briffa, K.; Osborn, T.; Tett, S. Two-hundred-fifty years of reconstructed and modeled tropical temperatures. J. Geophys. Res. Oceans 2006, 111, 1–13.

I hope these suggestions help enhance the clarity and impact of your manuscript.

ANS: Thank you. We also think that the text is now clearer.

References

Fritts, H., 1976. Tree rings and climate. The Blackburn Press, New Jersey.

Neukom, R., Steiger, N., Gómez-Navarro, J.J., Wang, J., Werner, J.P., 2019. No evidence for globally coherent warm and cold periods over the preindustrial Common Era. Nature 571, 550-554.

Tardif, R., Hakim, G.J., Perkins, W.A., Horlick, K.A., Erb, M.P., Emile-Geay, J., Anderson, D.M., Steig, E.J., Noone, D., 2019. Last millennium reanalysis with an expanded proxy database and seasonal proxy modeling. Clim. Past 15, 12